# Licorice Extract Isoliquiritigenin Protects Endothelial Function in Type 2 Diabetic Mice

**DOI:** 10.3390/nu16183160

**Published:** 2024-09-19

**Authors:** Lin Wang, Ruiwen Zhu, Chufeng He, Huixian Li, Qile Zhang, Yiu Ming Cheung, Fung Ping Leung, Wing Tak Wong

**Affiliations:** 1School of Life Sciences, The Chinese University of Hong Kong, Hong Kong, China; 1155136080@link.cuhk.edu.hk (L.W.); zhuruiwen@link.cuhk.edu.hk (R.Z.); maple-hecf@link.cuhk.edu.hk (C.H.); lihuixian@link.cuhk.edu.hk (H.L.); zhangqile@link.cuhk.edu.hk (Q.Z.); 1155092663@link.cuhk.edu.hk (Y.M.C.); christina.leung.2009@gmail.com (F.P.L.); 2Shenzhen Research Institute, The Chinese University of Hong Kong, Shenzhen 518172, China; 3State Key Laboratory of Agrobiotechnology, The Chinese University of Hong Kong, Hong Kong, China

**Keywords:** isoliquiritigenin, type 2 diabetes, endothelial dysfunction, interleukin-1β, reactive oxygen species, inflammatory factors

## Abstract

Endothelial dysfunction occurs prior to atherosclerosis, which is an independent predictor of cardiovascular diseases (CVDs). Diabetes mellitus impairs endothelial function by triggering oxidative stress and inflammation in vascular tissues. Isoliquiritigenin (ISL), one of the major bioactive ingredients extracted from licorice, has been reported to inhibit inflammation and oxidative stress. However, the therapeutic effects of ISL on ameliorating type 2 diabetes (T2D)-associated endothelial dysfunction remain unknown. In our animal study, *db*/*db* male mice were utilized as a model for T2D-associated endothelial dysfunction, while their counterpart, heterozygote *db*/*m*^+^ male mice, served as the control. Mouse brain microvascular endothelial cells (mBMECs) were used for in vitro experiments. Interleukin-1β (IL-1β) was used to induce endothelial cell dysfunction. ISL significantly reversed the impairment of endothelium-dependent relaxations (EDRs) in *db*/*db* mouse aortas. ISL treatment decreased ROS (reactive oxygen species) levels in *db*/*db* mice aortic sections and IL-1β-treated endothelial cells. Encouragingly, ISL attenuated the overexpression of pro-inflammatory factors MCP-1, TNF-α, and IL-6 in *db*/*db* mouse aortas and IL-1β-impaired endothelial cells. The NOX2 (NADPH oxidase 2) overexpression was inhibited by ISL treatment. Notably, ISL treatment restored the expression levels of IL-10, SOD1, Nrf2, and HO-1 in *db*/*db* mouse aortas and IL-1β-impaired endothelial cells. This study illustrates, for the first time, that ISL attenuates endothelial dysfunction in T2D mice, offering new insights into the pharmacological effects of ISL. Our findings demonstrate the potential of ISL as a promising therapeutic agent for the treatment of vascular diseases, paving the way for the further exploration of novel vascular therapies.

## 1. Introduction

Cardiovascular diseases (CVDs) include disorders of the heart and blood vessels [1]. It has been reported that CVDs are the leading cause of death worldwide [1]. Endothelium is an active inner layer of the blood vessels, which plays an important role in the maintenance of vascular homeostasis [2,3]. Chronic exposure to risk factors such as hyperglycemia, inflammation, and oxidative stress profoundly impair vascular homeostasis, ultimately leading to endothelial dysfunction [4]. Endothelial dysfunction is the initial step of vascular diseases. Preventing endothelial dysfunction is reported to be a promising therapeutic strategy to decrease cardiovascular deaths [5]. Endothelial dysfunction is characterized by the impairment of endothelium-dependent relaxations (EDRs), which is caused by the imbalance between endothelium-derived relaxing factors and endothelium-derived contracting factors [6,7].

Diabetes is a chronic metabolic disease triggered by high blood glucose levels, leading to serious damage to the heart, blood vessels, eyes, kidneys, and nerves. Diabetes is divided into three types: type 1 diabetes, type 2 diabetes, and gestational diabetes. It is reported that the vast majority of diabetes incidents are classified as type 2 diabetes (T2D) [8], which is characterized by the impairments of insulin secretion and insulin resistance [9]. Insulin resistance has been proven to impair endothelial function in T2D patients [2]. In addition, hyperglycemia triggers inflammation and oxidative stress in vascular tissues, leading to endothelial dysfunction [5,10]. Notably, the morbidity of cardiovascular diseases is elevated in individuals with T2D [11]. However, existing therapeutic strategies are inadequate to alleviate endothelial dysfunction in T2D patients [12,13]. Therefore, it is of the utmost importance to develop effective therapeutic treatments to ameliorate endothelial dysfunction in diabetic individuals. Attenuating inflammation and oxidative stress provide a promising direction for the development of novel vascular therapies [14,15].

Licorice, also known as Glycyrrhiza, has been served as a traditional Chinese medicine for over two thousand years [16]. The pharmacological effects of licorice on treating various diseases, such as lung injuries and heart diseases, have been demonstrated over the past four thousand years [17,18]. Isoliquiritigenin (ISL) is one of the most important bioactive compounds in licorice [19], which has been reported to exhibit anti-inflammatory properties via inhibiting NOD-like receptor protein 3 (NLRP3)/the nuclear factor kappa-light-chain-enhancer of the activated B (NF-κB) pathway in carrageenan-induced pleurisy [20]. Previous research has demonstrated that ISL alleviates inflammation by reducing the production of interleukin-1β (IL-1β)/caspase-1 [21]. In addition, ISL protects against traumatic brain injury via activating the nuclear factor erythroid-2-related factor 2 (Nrf2)/antioxidant responsive element (ARE) signaling pathway, highlighting the antioxidant value of ISL [22]. Moreover, other pharmacological activities of ISL, such as anti-cancer and anti-microbial properties, have been reported in previous studies, indicating the potential of ISL in treating human diseases [23]. In addition, ISL has been reported to alleviate insulin resistance and restore metabolic homeostasis in HFD-induced diabetic mice [24], while the mechanism underlying this antidiabetic effect involves the activation of 5′ adenosine monophosphate-activated protein kinase (AMPK) and the inhibition of the mammalian target of rapamycin complex 1 (mTORC1) [24]. In addition, ISL also has been reported to have the potential to treat complications of diabetes. ISL exhibits significant pharmacological effects on the deterioration of diabetic nephropathy by reducing inflammation and oxidative stress associated with its direct binding to Sirtuin 1 (SIRT1), the inhibition of mitogen-activated protein kinase (MAPK), and the induction of Nrf-2 signaling [25]. In diabetic cardiomyopathy, ISL effectively inhibits high glucose-induced hypertrophy, fibrosis, and apoptosis by alleviating the inflammatory response and oxidative stress via the inhibition of MAPK and induction of Nrf-2 signaling [26]. However, the effects of ISL on endothelial dysfunction, especially diabetes-induced endothelial dysfunction, remain largely unknown.

Hence, this study aims to investigate the effects of ISL on endothelial dysfunction in T2D mice. This study demonstrates that ISL reverses the impairment of endothelium-dependent relaxations (EDRs) in T2D mouse aortas. Furthermore, ISL attenuates oxidative stress and inflammation in T2D mouse aortas and IL-1β-treated endothelial cells. This study represents the first long-term investigation to evaluate the alleviative effects of ISL on the endothelial dysfunction associated with T2D, providing new insights into the therapeutic potential of ISL on vascular diseases.

## 2. Materials and Methods

### 2.1. Drugs and Solutions

Isoliquiritigenin was purchased from INDOFINE Chemical Company (Cat No. T-503, Hillsborough, NJ, USA). For ex vivo and in vitro studies, ISL was dissolved in Diemthyl Sulfoxide (DMSO, Cat No. 2206-27-1, Sigma-Aldrich Chemical, St. Louis, MO, USA). For in vivo studies, ISL was dissolved in PBS (Cat No. 10010072, ThermoFisher, Waltham, MA, USA). IL-1β was purchased from PeproTech (Cat No. 211-11B, Rocky Hill, NJ, USA). Dihydroethidium (DHE) was purchased from ThermoFisher (Cat No. D23107, Waltham, MA, USA). Acetylcholine (ACh, Cat No. A662), L-NG-nitro-L-arginine methyl ester (L-NAME, Cat No. N5501), phenylephrine (Phe, Cat No. P6126), and sodium nitroprusside (SNP, Cat No. 71778) were purchased from Sigma-Aldrich Chemical (St. Louis, MO, USA), and were dissolved in UltraPure™ DNase/RNase-Free Distilled Water (Cat No. 10977023, ThermoFisher, Waltham, MA, USA). The Krebs solution contained the following: (mmol/L) 119 NaCl, 4.7 KCl, 2.5 CaCl_2_, 1 MgCl_2_, 25 NaHCO_3_, 1.2 KH_2_PO_4_, and 11 D-glucose.

### 2.2. Animal Models

All animal procedures were performed following the Animal Experimentation Ethics Committee of the Chinese University of Hong Kong (CUHK) under Ref No. 22–226-MIS. *db*/*db* mice is a kind of T2D model lacking the gene encoding for leptin receptors from the C57BL/6J background. It has been reported that *db*/*db* mice exhibit key features of human T2D [27]. In this study, male *db*/*db* mice were used for the establishment of the T2D model. Counterpart heterozygote *db*/*m*^+^ mice were used as a vehicle control. All the animals were supplied by the Laboratory Animal Service Center of CUHK. The mice were kept in a temperature-controlled holding room (22–24 °C) with a 12 h light/dark cycle. A standard chow diet and water were provided for free. All the cages of mice were kept on the same shelf in the animal house. Both the research team (daily) and the veterinary staff (weekly) monitored the mice’s health. Health was monitored by weight (weekly), food intake (weekly), and the general assessment of animal activity, panting, and fur condition. Mice with more than 20% weight loss or obvious distress were euthanized. All *db*/*db* mice had established blood glucose levels over 33.3 mmol/dL before they were randomly divided into 2 groups. For the in vivo experiments, ten-week-old *db*/*db* and *db*/*m*^+^ mice were treated with ISL (20 mg/kg body weight/day, dissolved in PBS) for 8 weeks via oral gavage. An equal volume of PBS was given as the vehicle control. The mice received oral gavage treatment at the same time each day in a random order. Ten-week-old *db*/*db* and *db*/*m*^+^ mice were used for the ex vivo experiments. Based on a power calculation (estimated effect 60%, standard deviation of 5%, a power of 0.9, and alpha of 0.05) [28] and literature review [24], the sample size of each group was 6 mice for both the in vivo and the ex vivo experiments. There was no exclusion of any animals included in each group in this study.

### 2.3. Measurements of Basic Parameters

The body weight and food intake were recorded every week. The oral glucose tolerance test (OGTT) was conducted before the mice were sacrificed. The mice received a subcutaneous injection of Temgesic (0.05 mg/kg body weight, supplied by the Animal Experimentation Ethics Committee of the CUHK) to relieve pain during the OGTT. Blood was collected into heparinized tubes from the orbital sinus after anesthetizing the mice before euthanasia. The serum levels of blood glucose were detected using the Glucose LiquiColor^®^ test kit (Cat No. 1070-500, Stanbio, Boerne, TX, USA). The serum levels of total cholesterol were detected by the Cholesterol LiquiColor^®^ test kit (Cat No. 1010-225, Stanbio, Boerne, TX, USA). The serum levels of triglyceride were detected by the LiquiColor^®^ Triglycerides test kit (Cat No. 2200-225, Stanbio, Boerne, TX, USA).

### 2.4. Mice Arterial Preparation and Vascular Reactivity Study Determined by Wire Myograph

The isolated aortas were dissected and cleaned of adhering connective tissues. Afterward, the aortas were cut into ring segments (~2 mm in length) to detect vascular reactivity in the in vivo experiments. In the ex vivo studies, the aortic rings were cultured in Dulbecco’s Modified Eagle’s Medium (DMEM; 1 g/L glucose, Cat No. 11885084, Gibco, Gaithersburg, MD, USA) with or without 5 μmol/L ISL for 16 h. After specific treatments, the cultured aortic rings were mounted on the wire myograph system (620 M, DMT, Aarhus, Denmark) containing oxygenated (95% O_2_ and 5% CO_2_) Krebs solution to detect vascular reactivity. Changes in the isometric tone of the aortic rings were recorded. The aortic rings were stretched to an optimal baseline tension of 3 mN and then equilibrated for 60 min in Krebs solution under 37 °C. The aortic rings were precontracted with 60 mmol/L KCl for 15 min and then washed with Krebs solution. After several washouts, phenylephrine (Phe, 3 μmol/L) was used to induce the contraction. Acetylcholine (ACh, 3 nmol/L to 10 μmol/L) was added cumulatively to induce endothelium-dependent relaxations. Afterward, the aortic rings were incubated with L-NG-nitro-L-arginine methyl ester (L-NAME) for 15 min to inhibit the production of nitric oxide (NO) in vascular endothelium. Endothelium-independent relaxations were determined by cumulatively adding sodium nitroprusside (SNP, 1 nmol/L to 10 μmol/L).

### 2.5. Cell Culture

Mouse brain microvascular endothelial cells (mBMECs) were purchased from Angio-proteomie (Cat No. cap-m0002-GFP, Boston, MA, USA). The cells were cultured in a DMEM high glucose medium (4.5 g/L glucose, Cat No. 11965118, Gibco, Gaithersburg, MD, USA) supplemented with 10% fetal bovine serum (FBS, Gibco, Cat No. 10270106, Grand Island, NE, USA) and 1% Antibiotic-Antimycotic (Cat No. 15240096, Gibco, Gaithersburg, MD, USA). The cells were maintained at 37 °C with 95% O_2_ and 5% CO_2_. Cells were treated with 1 ng/mL IL-1β [15,29,30] with or without 5 μmol/L ISL for 16 h. There was no exclusion in any of the groups.

### 2.6. Reactive Oxygen Species (ROS) Measurement

The aortic rings were embedded by an optimal cutting temperature compound (O.C.T., Tissue-Tek^®^ O.C.T.™ Compound, Sakura Finetek Europe B.V., Alphen aan den Rijn, The Netherland) and were then frozen using liquid nitrogen. The embedded aortic rings were sectioned into 6 μm thick slides. The mBMECs were seeded in confocal dishes (Cat No. 200350, SPL Life Sciences, Gyeonggido, Republic of Korea) with a DMEM high glucose medium (4.5 g/L glucose, Cat No. 11965118, Gibco, Gaithersburg, MD, USA). Cells were treated with 1 ng/mL IL-1β and/or 5 μmol/L ISL for 16 h before ROS detection. Both aortic rings and cells were stained with 5 μmol/L dihydroethidium (DHE, Cat No. D23107, Invitrogen, Waltham, MA, USA) for 30 min at room temperature. DHE intensity was detected by confocal microscope (Excitation 518 nm, Emission 606 nm, TCS SP8 MP, Leica, Wetzlar, Germany). The intensity was analyzed by Image J (Version 1.52a, National Institutes of Health, Bethesda, MD, USA).

### 2.7. Quantitative Polymerase Chain Reaction (qPCR) Analysis

Total RNA was extracted from mBMECs and mouse aortas with RNAiso plus reagents (Cat No. 9109, Takara, Shiga, Japan). A spectrophotometer (NanoDrop One, Thermo Fisher Scientific, Tokyo, Japan) was used to determine the RNA concentration. Afterward, 1000 ng of RNA was reversed into cDNA using the PrimeScript™ MasterMix kit (Cat No. RR036B, Takara, Shiga, Japan). Quantitative polymerase chain reaction (qPCR) experiments were then performed using the qPCR system (BioRad, CFX96, Hercules, CA, USA) with TB Green^®^ Premix Ex Taq™ kit (Cat No. RR420A, Takara, Japan) according to the manufacturer’s protocol. mRNA expression levels were normalized to the housekeeping gene β-actin. The delta–delta CT method was used to analyze the results.

### 2.8. Statistical Analysis

Results were presented as the mean ± standard error of the mean (SEM). For the animal study, the n value was six mice. For the in vitro study, the n value was five independent experiments. Comparisons among different groups were performed using one-way or two-way ANOVA followed by Tukey’s test. Data were analyzed using GraphPad Prism software (Version 9.0, San Diego, CA, USA). *p* < 0.05 was regarded as statistically significant.

For the blinding procedure, at the allocation stage, the experimenters (Huixian Li and Qile Zhang) numbered the sacrifice order randomly. During the conduction of the experiments, the experimenters (Lin Wang, Ruiwen Zhu, Chufeng He) only knew the sacrifice order and did not know which groups the animals belonged to. After sacrifice and data collection, all the data were reorganized by the experimenters responsible for determining the sacrifice order (Huixian Li and Qile Zhang) according to the experimental group. During the data analysis stage, the experimenters (Lin Wang, Ruiwen Zhu, Chufeng He, You Ming Cheung, and Fung Ping Leung) only knew which groups the data belonged to and did not know the sacrifice order.

## 3. Results

### 3.1. ISL Oral Treatment Reverses Endothelial Dysfunction in db/db Mouse Aortas

We first investigated whether ISL oral treatment improves endothelium-dependent relaxations (EDRs) in *db*/*db* mice (Figure 1A). The acetylcholine (ACh)-induced EDRs were impaired in *db*/*db* mice compared to the *db*/*m*^+^ counterparts. By contrast, ISL treatment improved the EDRs in *db*/*db* mouse aortas (Figure 1B,C), demonstrating that ISL attenuates endothelial dysfunction in T2D mice. The endothelium-independent relaxations induced by sodium nitroprusside (SNP) were similar among the groups (Figure 1D), indicating that ISL may not significantly affect the function of the smooth muscles in T2D mice. In addition, the mRNA expression levels of endothelial activation markers, VCAM-1 and ICAM-1, were upregulated in *db*/*db* mouse aortas. Encouragingly, ISL treatment significantly decreased the expression levels of VCAM-1 and ICAM-1 in *db*/*db* mouse aortas (Figure 1E,F), further demonstrating that ISL reverses endothelial dysfunction in T2D mouse aortas. Taken together, the above findings demonstrate that ISL ameliorates endothelial dysfunction in *db*/*db* mouse aortas without affecting endothelium-independent relaxations.

### 3.2. ISL Reverses the Impairment of Endothelium-Dependent Relaxations (EDRs) in db/db Mouse Aortas in Ex Vivo Studies

We next examined whether ISL treatment improves vascular function by conducting ex vivo experiments to further explore the therapeutic effects of ISL on endothelial dysfunction. The ACh-induced EDRs were impaired in the aortas derived from *db*/*db* mice. Importantly, ISL treatment restored the impairments of EDRs in *db*/*db* mice (Figure 1G), demonstrating that ISL attenuates the endothelial dysfunction induced by T2D. SNP-induced endothelium-independent relaxations were similar among the groups (Figure 1H). These findings collectively illustrate that ISL reverses EDRs in *db*/*db* mouse aortas without affecting endothelium-independent relaxations, which is consistent with our previous in vivo observations.

### 3.3. Basic Parameters of Mice Treated with ISL

As a well-known metabolic disease, T2D affects basic parameters, including blood glucose levels, blood lipid levels, body weight, and fat mass, ultimately leading to endothelial dysfunction [31]. Hence, we evaluated the effects of ISL on the general metabolic health of *db*/*db* mice. The results revealed that ISL treatment did not affect body weight gain (Appendix A) or food intake (Appendix A) in *db*/*db* mice. In addition, the oral glucose tolerance test showed that the blood glucose levels of *db*/*db* mice failed to decrease back to the starting level after 2 h of glucose solution administration, indicating that *db*/*db* mice suffered from insulin resistance, which is the hallmark of T2D. Notably, ISL treatment did not ameliorate insulin resistance in *db*/*db* mice (Appendix A). Furthermore, the percentage of different types of adipose tissues in body weight was not decreased in ISL-treated *db*/*db* mice. The percentages of subcutaneous fat (SCF), perigenital fat (PGF), and perirenal fat (PRF) in body weight significantly increased in *db*/*db* mice, while ISL treatment did not decrease the percentage of SCF, PGF, and PRF in body weight in *db*/*db* mice (Appendix A). The percentage of perivascular adipose tissue (PVAT) and brown adipose (BAT) tissue remained similar among these groups (Appendix A). In addition, ISL did not affect the levels of plasma glucose (Appendix A), plasma cholesterol (Appendix A), or plasma triglyceride (Appendix A) in *db*/*db* mice. In summary, the above findings demonstrate that ISL exhibits protective effects on vascular health without affecting blood glucose levels, blood lipid levels, glucose tolerance, or fat mass.

### 3.4. Oral Treatment of ISL Attenuates Oxidative Stress in db/db Mice

Since oxidative stress is an important contributor to endothelial dysfunction in T2D [31], we next further investigated whether ISL attenuates vascular oxidative stress in *db*/*db* mice. Dihydroethidium (DHE) staining results reflected that the total levels of reactive oxygen species (ROS) increased in *db*/*db* mice aortic rings, indicating that oxidative stress occurs in *db*/*db* mouse aortas. Notably, oral ISL treatment effectively reduces ROS levels in *db*/*db* mouse aortas (Figure 2A,B), demonstrating that ISL attenuates vascular oxidative stress in *db*/*db* mouse aortas. Quantitative PCR (qPCR) results showed that the expression level of NOX2, which is responsible for producing cellular ROS [32], is upregulated in *db*/*db* mouse aortas. However, ISL treatment significantly suppressed NOX2 overexpression in *db*/*db* mice (Figure 2C). Moreover, ISL treatment significantly restored the mRNA levels of SOD1 (Figure 2D), Nrf2 (Figure 2E), and HO-1 (Figure 2F) in *db*/*db* mouse aortas, highlighting the antioxidant value of ISL. To conclude, the above findings demonstrate that ISL attenuates oxidative stress in *db*/*db* mouse aortas. Specifically, ISL inhibits NOX2 overexpression and upregulates the mRNA expression of SOD1, Nrf2, and HO-1 in *db*/*db* mouse aortas.

### 3.5. ISL Reduces ROS Levels in db/db Mouse Aortas in Ex Vivo Studies

We next conducted ex vivo studies to further investigate whether ISL scavenges excessive ROS in isolated *db*/*db* mouse aortas. DHE staining results revealed that ISL incubation significantly reduces ROS levels in *db*/*db* mouse aortas (Figure 3A,B), which is consistent with our previous in vivo results. Taken together, the above findings demonstrate that ISL suppresses vascular oxidative stress in T2D mice.

### 3.6. ISL Alleviates Oxidative Stress Triggered by IL-1β in Endothelial Cells

To gain precise insights into the therapeutic effects of ISL in reversing the endothelial dysfunction associated with T2D, in vitro studies were introduced to further examine our previous conclusions. Since interleukin-1β (IL-1β) has been reported as a key player in the development of T2D and diabetic cardiovascular complications, we employed IL-1β to induce endothelial cell dysfunction in the following in vitro studies [31,33,34]. Our results showed that ISL incubation reduced heightened ROS levels, which were induced by IL-1β, in endothelial cells (Figure 3C,D), which is consistent with the previous in vivo observations. Moreover, IL-1β-upregulated NOX2 was reduced by ISL treatment (Figure 3E). In addition, the IL-1β-downregulated expression levels of SOD1 (Figure 3F), Nrf2 (Figure 3G), and HO-1 (Figure 3H) were restored in ISL-treated endothelial cells. In conclusion, these above findings demonstrate that ISL treatment attenuates oxidative stress in endothelial cells.

### 3.7. ISL Attenuates Inflammation in db/db Mouse Aortas and Endothelial Cells

Inflammation is another important inducer of endothelial dysfunction in T2D and ultimately leads to atherosclerosis [31]. mRNA expression levels of inflammatory cytokines were detected in both in vivo and in vitro experiments. The results showed that ISL treatment significantly inhibited the overexpression of pro-inflammatory cytokines, including MCP-1 (Figure 4A), TNF-α (Figure 4B), and IL-6 (Figure 4C), in *db*/*db* mouse aortas. The decreased expression level of IL-10, an anti-inflammatory cytokine, was upregulated in ISL-treated *db*/*db* mouse aortas (Figure 4D). These findings demonstrate that ISL exhibits anti-inflammatory properties in *db*/*db* mouse. We also conducted in vitro studies to further validate whether ISL alleviates inflammation in endothelial cells. Likewise, the overexpression of MCP-1 (Figure 4E), TNF-α (Figure 4F), and IL-6 (Figure 4G) was observed in IL-1β-treated endothelial cells. Encouragingly, ISL treatment remarkably suppresses the overexpression of MCP-1 (Figure 4E), TNF-α (Figure 4F), and IL-6 (Figure 4G) in IL-1β-impaired endothelial cells. Moreover, the IL-1β-decreased mRNA expression of IL-10 was significantly restored by ISL treatment (Figure 4H), which is consistent with the previous in vivo studies. Overall, these above findings collectively demonstrate that ISL alleviates inflammation in *db*/*db* mouse aortas and endothelial cells.

## 4. Discussion

Our study demonstrates, for the first time, the potential of ISL to ameliorate endothelial dysfunction in T2D mice. Specifically, ISL reverses the impairment of EDRs in T2D mice, which is the first long-term in vivo study to investigate the protective effects of ISL on vascular endothelium. Furthermore, our results demonstrate that ISL decreases ROS levels, downregulates NOX2, and upregulates antioxidant gene expressions in T2D mouse aortas and IL-1β-treated endothelial cells. Moreover, ISL decreases the expression levels of pro-inflammatory cytokines and significantly restores the mRNA expression of IL-10 in T2D mouse aortas and IL-1β-treated endothelial cells.

Endothelial dysfunction is a promising therapeutic target for reducing cardiovascular risks [5]. Endothelial dysfunction contributes to the pathogenesis of CVDs by promoting inflammation, oxidative stress, and thrombosis in the vasculature, leading to the impairment of the arterial tone [31]. Impaired endothelium-dependent relaxations (EDRs) are one of the key features of endothelial dysfunction [6,7]. Hyperglycemia triggers low-grade systemic inflammation and oxidative stress, thereby leading to endothelial dysfunction and eventually promoting the progression of CVDs [5,10,31]. However, current therapeutic strategies are inadequate to attenuate endothelial dysfunction and vasculopathy progression, particularly in diabetic patients [12,13]. Therefore, it is of the utmost importance to develop effective treatments to ameliorate endothelial dysfunction in diabetic individuals [14,15,31]. In our study, we evaluated the therapeutic effects of ISL on ameliorating endothelial dysfunction in the T2D model. Promisingly, both in vivo and ex vivo results collectively demonstrate that ISL reverses impaired EDRs in *db*/*db* mice. Meanwhile, the increased mRNA expression levels of endothelial dysfunction markers, VCAM-1 and ICAM-1, were decreased in ISL-treated *db*/*db* mouse aortas, further confirming that ISL reverses T2D-induced endothelial dysfunction. In addition, our results showed that oral ISL treatment did not affect blood glucose levels, blood lipid levels, body mass, and insulin resistance in T2D mice. These findings indicate that ISL prevents vascular endothelium from hyperglycemia damage without affecting metabolic parameters. Existing therapeutic strategies to alleviate T2D and T2D-associated endothelial dysfunction mainly through attenuating hyperglycemia, lowering blood lipids, and alleviating insulin resistance [15,35,36]. Based on our findings, further investigations that delve into the underlying mechanisms of ISL in ameliorating T2D-induced endothelial dysfunction can pave the way for the further development of novel vascular therapies. Furthermore, this study demonstrates that ISL treatment reverses the impairment of endothelium-dependent relaxations without affecting the function of smooth muscle cells, which indicates that ISL may alleviate vascular dysfunction by targeting endothelium. These findings point out the direction for future elucidations of the mechanism by which ISL improves T2D-associated endothelial dysfunction, contributing to the further exploration and establishment of ISL as a vascular agent and dietary supplement for diabetic individuals.

Chronic exposure to hyperglycemia induces inflammation and oxidative stress in vascular tissues, which are two well-known risk factors leading to endothelial dysfunction in T2D [14,31,37]. Endothelial dysfunction is characterized by the impairment of EDRs, which is caused by the decreased production and bioavailability of NO [5]. In this study, we also examined whether ISL alleviates inflammation and oxidative stress in *db*/*db* mouse aortas and endothelial cells. The results revealed that ISL treatment significantly suppresses inflammatory responses and excessive ROS levels in aortic sections and IL-1β-impaired endothelial cells. Specifically, ISL treatment remarkably suppresses the overexpression of pro-inflammatory cytokines, MCP-1, TNF-α, and IL-6, in *db*/*db* mouse aortas. The decreased expression level of IL-10 was reversed by ISL treatment. These above findings were likewise reflected in in vitro studies. Furthermore, oral ISL treatment reduced elevated ROS levels in *db*/*db* mouse aortas. SOD1 is a pivotal enzyme involved in the degradation of superoxide [38]. Encouragingly, ISL treatment significantly increased SOD1 expression in *db*/*db* mouse aortas. Nrf2 and HO-1 are two well-known factors to clear excessive intracellular ROS [39]. Notably, the downregulated expressions of Nrf2 and HO-1 were restored by ISL treatment in *db*/*db* mouse aortas. NOX2 is one of the major sources of cellular ROS in vascular tissues [32]. Impressively, ISL treatment decreased NOX2 expression in *db*/*db* mouse aortas. These above findings were also likewise reflected in in vitro studies, highlighting the antioxidant value of ISL in vascular endothelium. Taken together, these findings first demonstrate that ISL attenuates vascular inflammation and oxidative stress, contributing to ameliorating the endothelial dysfunction associated with T2D. Numerous studies have demonstrated the high impact of hyperglycemia-induced endothelial injuries on other complications of diabetes, including diabetic hepatopathy and diabetic cardiomyopathy [40,41]. Our above findings prompt us to further explore the therapeutic potential of ISL in alleviating other diabetes complications. These investigations not only provide new insights into the pharmacological effects of ISL but also offer scientific validation for the further establishment of ISL as a therapeutic agent for the treatment of T2D and T2D-related complications.

This study represents the first long-term investigation that evaluates the protective effects of ISL on endothelial dysfunction associated with T2D. However, the molecular mechanism by which ISL ameliorates endothelial dysfunction remains to be further investigated. Furthermore, this study did not examine all the inducing factors of endothelial dysfunction in T2D. Therefore, further investigations can be conducted to characterize whether ISL treatment affects other potential risk factors leading to T2D-associated endothelial dysfunction. These elucidations provide novel insights into the therapeutic effects of ISL on vascular dysfunction. In addition, it introduces a comparison with other bioactive antidiabetic flavonoids, such as quercetin [42,43], luteolin [42,43], apigenin [43], and genistein [42,43], on the therapeutic effects of T2D-associated endothelial dysfunction, is also an important direction for the further investigation of the therapeutic potential of ISL.

## 5. Conclusions

Our results demonstrate that ISL reverses the impairment of endothelium-dependent relaxations in T2D mice. Moreover, ISL alleviates inflammation and oxidative stress in *db*/*db* mouse aortas and IL-1β-treated endothelial cells. These findings highlight the potential of ISL as a promising vascular agent for the treatment of vasculopathy, providing a new vascular therapy for diabetic patients.

## Figures and Tables

**Figure 1 nutrients-16-03160-f001:**
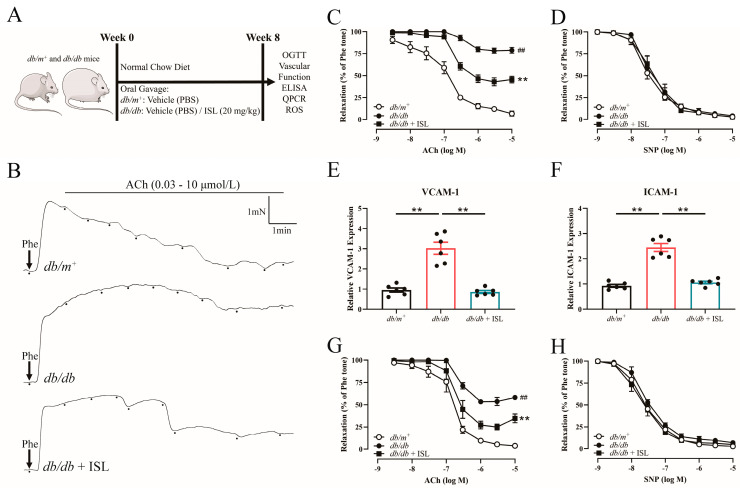
Isoliquiritigenin (ISL) treatment reverses endothelial dysfunction in type 2 diabetes mouse aortas. (**A**) Experimental design of the animal study. (**B**) Representative tracings and (**C**) summarized results showing the beneficial effects of ISL treatment on impaired acetylcholine-induced endothelium-dependent relaxations (EDRs) in *db*/*db* mice. (**D**) Endothelium-independent relaxations in response to sodium nitroprusside (SNP) remain similar among different groups. (**E**,**F**) qPCR analysis demonstrates that ISL treatment downregulates mRNA expression levels of VCAM-1 (**E**) and ICAM-1 (**F**) in *db*/*db* mice. (**G**,**H**) Aortas isolated from *db*/*m*^+^ and *db*/*db* mice were treated with 5 μmol/L ISL for 16 h. (**G**) Summarized results showing the beneficial effects of ISL treatment on impaired acetylcholine-induced EDRs in *db*/*db* mice. (**H**) Endothelium-independent relaxations in response to SNP remained similar among different groups. Data represent means ± SEM of 6 mice. mRNA expressions were normalized with β-actin. *p* values were determined using two-way ANOVA for (**C**,**D**,**G**,**H**) and one-way ANOVA for (**E**,**F**). ^##^
*p* < 0.0001 *db*/*db* vs. *db*/*m*^+^, ** *p* < 0.01 *db*/*db* + ISL vs. *db*/*db* (C,D,G,H). ** *p* < 0.01 (**E**,**F**). ISL, isoliquiritigenin; OGTT, oral glucose tolerance test; qPCR, quantitative polymerase chain reaction; ROS, reactive oxygen species; ELISA, enzyme-linked immunosorbent assay; ACh, acetylcholine; SNP, sodium nitroprusside; VCAM-1, vascular cell adhesion molecule 1; and ICAM-1, intercellular adhesion molecule 1.

**Figure 2 nutrients-16-03160-f002:**
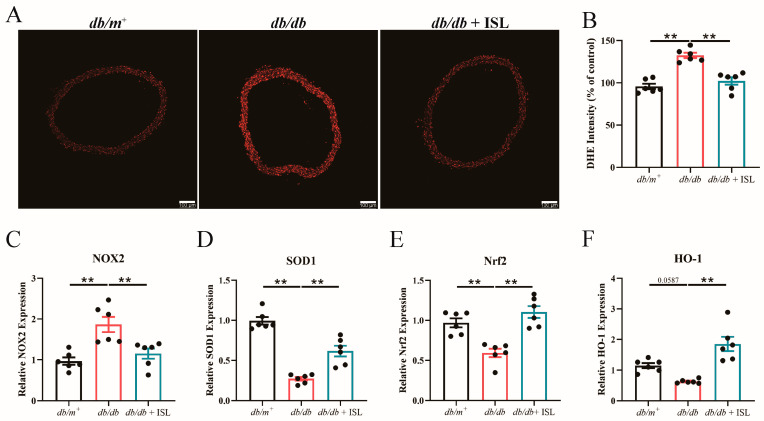
Oral ISL treatment attenuates oxidative stress in *db*/*db* mouse aortas. *db*/*db* mice were treated with 20 mg/kg of ISL via oral gavage for 8 weeks. (**A**) Representative images and (**B**) summarized results of DHE staining demonstrate that ROS levels increase in *db*/*db* mouse aortas. ISL treatment decreases ROS levels in *db*/*db* mouse aortas. (**C**–**F**) qPCR analysis demonstrates that ISL treatment downregulates the mRNA expression level of NOX2 (**C**) in *db*/*db* mouse aortas. ISL upregulates mRNA expression levels of SOD1 (**D**), Nrf2 (**E**), and HO-1 (**F**) in *db*/*db* mouse aortas. Scale bar: 100 μm. Data represent the means ± SEM of 6 mice. mRNA expressions were normalized with β-actin. *p* values were determined using one-way ANOVA (**B**–**F**). ** *p* < 0.01. ISL, isoliquiritigenin; DHE, dihydroethidium; NOX2, NADPH oxidase 2; SOD1, superoxide dismutase 1; Nrf2, nuclear factor erythroid-2 related factor 2; and HO-1, heme oxygenase-1.

**Figure 3 nutrients-16-03160-f003:**
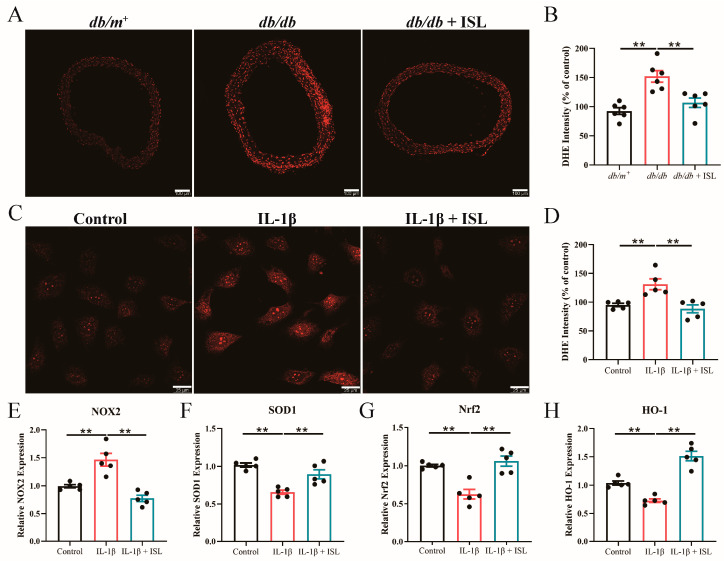
ISL suppresses oxidative stress in isolated *db*/*db* mouse aortas and IL-1β-treated endothelial cells. (**A**,**B**) Aortas isolated from *db*/*m*^+^ and *db*/*db* mice were treated with 5 μmol/L ISL for 16 h. (**A**) Representative images and (**B**) summarized results of DHE staining demonstrate that ISL decreases enhanced ROS levels in *db*/*db* mouse aortas. (**C**,**H**) Endothelial cells were treated with IL-1β (1 ng/mL) and/or ISL (5 μmol/L) for 16 h. (**C**) Representative images and (**D**) summarized results of DHE staining indicate that ISL suppresses heightened ROS levels in IL-1β-treated endothelial cells. (**E**–**H**) qPCR analysis demonstrates that ISL treatment downregulates mRNA expression levels of NOX2 (**E**) in IL-1β-treated endothelial cells. ISL restores SOD1 (**F**), Nrf2 (**G**), and HO-1 (**H**) downregulation in IL-1β-treated endothelial cells. Scale bar: 100 μm (**A**), 25 μm (**C**). Data represent the means ± SEM of 6 mice (**A**,**B**) or 5 experiments (**C**–**H**). mRNA expressions were normalized with β-actin. *p* values were determined using one-way ANOVA for (**B**–**H**). ** *p* < 0.01. ISL, isoliquiritigenin; DHE, dihydroethidium; IL-1β, interleukin-1β; NOX2, NADPH oxidase 2; SOD1, superoxide dismutase 1; Nrf2, nuclear factor erythroid-2 related factor 2; and HO-1, heme oxygenase-1.

**Figure 4 nutrients-16-03160-f004:**
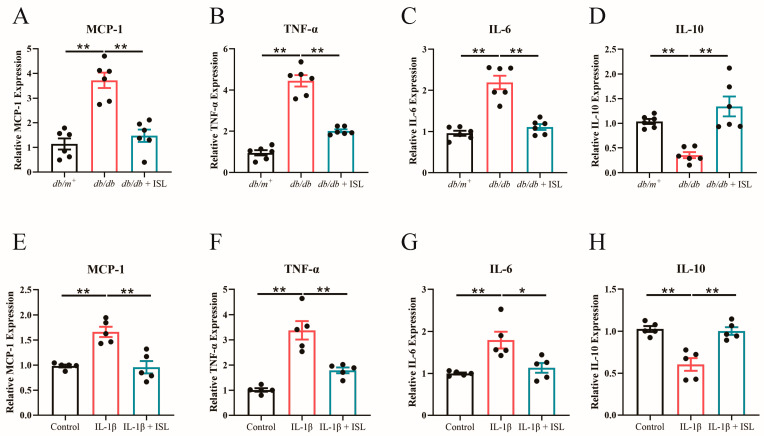
ISL attenuates inflammation in *db*/*db* mouse aortas and IL-1β-treated endothelial cells. (**A**–**D**) *db*/*db* mice were treated with 20 mg/kg of ISL via oral gavage for 8 weeks. qPCR analysis demonstrates that ISL treatment downregulates mRNA expression levels of MCP-1 (**A**), TNF-α (**B**), and IL-6 (**C**) in *db*/*db* mouse aortas. ISL upregulates the mRNA expression of IL-10 (**D**) in *db*/*db* mouse aortas. (**E**–**H**) Endothelial cells were treated with IL-1β (1 ng/mL) and/or ISL (5 μmol/L) for 16 h. qPCR analysis demonstrates that ISL treatment downregulates mRNA expression levels of MCP-1 (**E**), TNF-α (**F**), and IL-6 (**G**) in IL-1β-treated endothelial cells. ISL upregulates the mRNA expression of IL-10 (**H**) in IL-1β-endothelial cells. mRNA expressions were normalized to β-actin. Data represent the means ± SEM of 6 mice (**A**–**D**) or 5 experiments (**E**–**H**). *p* values were determined using one-way ANOVA. * *p* < 0.05, ** *p* < 0.01. ISL, isoliquiritigenin; IL-1β, interleukin-1β; MCP-1, monocyte chemoattractant protein 1; TNF-α, tumor necrosis factor-α; IL-6, interleukin-6; and IL-10, interleukin-10.

## Data Availability

Data are available with the permission of the corresponding author.

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
