# Peer review of "Licorice Extract Isoliquiritigenin Protects Endothelial Function in Type 2 Diabetic Mice"

_nutrients, 2024, doi:10.3390/nu16183160_

Round 1

Reviewer 1 Report

Comments and Suggestions for Authors

The authors report antidiabetic effects in vivo with the chalcone isoliquiritigenin that is found in licorice.

There are many previous studies of this type but they are not discussed or cited. On PubMed alone there are 40 publications investigating isoliquiritegenin in diabetes.

This manuscript confirms that the natural product has antidiabetic effects, which is already well known. However, it does not provide any new insight into the cellular targets of the natural product. 

The possibility of isoliquritigenin isomerizing to liquritigenin is not mentioned and it is unknown whether this happens during their in vivo conditions.

The experiments do not have a positive control, so we do not know if the expression changes are significant compared to other chalcones or flavonoids that are bioactive.

Reviewer 2 Report

Comments and Suggestions for Authors

This study is focused on the protective effects of licorice extract isoliquiritigenin (ISL) on endothelial function in type 2 diabetic (T2D) mice, which is an important and interesting topic of research due to the need to find therapeutic and prophylactic solutions for atherosclerosis and cardiovascular diseases (CVD) in patients with T2D. Please refer to the following observations:

Abstract- please define „NOS”;

Keywords-consider adding supplementary words, such as „interleukin-1β”, „reactive oxygen species”, and „inflammatory factors”, in order to improve the discoverability of the paper.

Introduction- maybe expanding a bit the section about the mechanisms of action of ISL in T2D would be beneficial for the readers- https://pubmed.ncbi.nlm.nih.gov/35114453/, https://www.ncbi.nlm.nih.gov/pmc/articles/PMC7721869/

Conformity with the ARRIVE guidelines:

-2a - I could not find it in section 2.2. any reference to the exact number of mice in each arm; only the footnotes of Fig.1-4 state that six mice were involved;

-3b - „If there were no exclusion, state so.” – please also insert this in the body text, not only in the table;

-3c- in section 2.8, there is no specific reference to the value of „n”;

-4b- what methods were used to minimize potential confounders?

-5- the description of the blinding procedure should be inserted in the body text, not only in the „Authors contribution” section; for example, Author 1 was aware of the group allocation, but not Author 2, who conducted the statistical analysis, and Authors 3 and 4 who conducted the product administration, etc.;

-6b- it is not clear how the sample size was determined;

-11- there is no reference to the sex of the mice in the „Abstract” section;

-12b- the explanations about the relevance to human biology are insufficient, please consider adding more data, or specify there are no data available;

-16a- there is insufficient data on this subject, please see details here- https://arriveguidelines.org/arrive-guidelines/animal-care-and-monitoring/16a/explanation

-16b- is there any reason why the Authors consider reporting adverse events was not applicable here?

-16c- there is virtually no data on „humane endpoints” in section 2.2- please see details on this subject here- https://arriveguidelines.org/arrive-guidelines/animal-care-and-monitoring/16c/explanation

Round 2

Reviewer 2 Report

Comments and Suggestions for Authors

The quality of the manuscript improved.

There are some formal aspects that needs to be revised, like the format of text in lines 182-190, placing the title 2.8 in a distinct line, placing the title 5 in a new line, and eliminating the highlighting of the text in rows 429-440.

Round 3

Reviewer 2 Report

Comments and Suggestions for Authors

The manuscript has substantially improved.